# Comprehensive Schottky Barrier Height Behavior and Reliability Instability with Ni/Au and Pt/Ti/Pt/Au on AlGaN/GaN High-Electron-Mobility Transistors

**DOI:** 10.3390/mi13010084

**Published:** 2022-01-04

**Authors:** Surajit Chakraborty, Tae-Woo Kim

**Affiliations:** Department of Electrical, Electronic and Computer Engineering, University of Ulsan, Ulsan 44610, Korea; surajit5103@ulsan.ac.kr

**Keywords:** AlGaN/GaN, critical voltage, degradation, off-state stress, inverse piezoelectric effect, high temperature, Schottky barrier height, reliability instability

## Abstract

The reliability instability of inhomogeneous Schottky contact behaviors of Ni/Au and Pt/Ti/Pt/Au gate contacts on AlGaN/GaN high-electron-mobility transistors (HEMTs) was investigated via off-state stress and temperature. Under the off-state stress condition, Pt/Ti/Pt/Au HEMT showed abruptly reduced reverse leakage current, which improved the Schottky barrier height (SBH) from 0.46 to 0.69 eV by suppression of the interfacial donor state. As the temperature increased, the reverse leakage current of the Pt/Ti/Pt/Au AlGaN/GaN HEMT at 308 K showed more reduction under the same off-state stress condition while that of the Ni/Au AlGaN/GaN HEMT increased. However, with temperatures exceeding 308 K under the same off-state stress conditions, the reverse leakage current of the Pt/Ti/Pt/Au AlGaN/GaN HEMT increases, which can be intensified using the inverse piezoelectric effect. Based on this phenomenon, the present work reveals the necessity for analyzing the concurrent SBH and reliability instability due to the interfacial trap states of the MS contacts.

## 1. Introduction

Gallium nitride (GaN)-based high-electron-mobility transistors (HEMTs) have attracted attention in microwave and power-switching applications due to their unparalleled properties including wide direct bandgap, high-temperature operation, high breakdown field, and high-saturation electron velocity [1,2,3]. Because of piezoelectric and spontaneous polarizations in AlGaN/GaN, GaN-based HEMTs have been used to achieve high carrier densities and high mobilities at the heterostructure without doping by the origination of two-dimensional electron gas (2DEG) channels [4,5,6]. The electrical performances of GaN-based HEMT devices show sturdy functions of both Schottky and Ohmic metal contacts because the Schottky contact in AlGaN/GaN is controlled by the current conduction in the channel [7]. The gate reverse leakage current still remains a key concern for GaN-HEMT stability due to the interfacial trap states and strain-induced defects such as dislocations and cracks from large lattice mismatches [8,9,10,11,12]. Severe limitations such as current collapse, power slump, and poor long-term reliability are induced by the trapping effects at the interfacial trap states, trap sites in the AlGaN barrier layer, and deep-level traps in the GaN buffer layer [13,14,15].

In particular, trapping characteristics were evaluated to reduce the density of states for the surface donors using different gate metals on the AlGaN/GaN heterostructure [16,17]. One of the solutions proposed for such GaN-based HEMTs is the improvement in the interactions between the metal and semiconductor surfaces mediated by dangling bonds [18]. Schottky contacts on GaN and AlGaN/GaN heterostructures have been investigated for various metals including Pt, Au, Ti, Ni, Au, and other highly heat resistant metals [19,20,21]. Despite the excellent tuning of the Schottky barrier height (SBH) on a GaN-based HEMT, the reliability instability remains a primary concern for device optimization due to the relationships between the qualities of the AlGaN/GaN epitaxial layers and their geometries [22]. In particular, the carriers in the GaN channel can achieve high energies at the gate edge on the drain side, where the combined injection velocity from the lateral and vertical directions is the maximum [23]. The degradation of the hot carriers may cause a significant reduction in the drain current and transconductance (Gm) as well as shifts in the threshold voltage (VT), resulting in decreased direct current (DC), radio frequency (RF), and large-signal performances [24,25]. To quantitatively understand the electrostatic characteristics of the AlGaN/GaN heterostructure, it is necessary to comprehensively analyze the concurrent SBH and reliability instability due to the interfacial trap states of the metal–semiconductor (MS) contacts.

In this study, we investigated the comprehensive SBH and temperature as well as device degradation of Ni/Au and Pt/Ti/Pt/Au contacts on AlGaN/GaN HEMTs. The Schottky behavior characteristics for the Ni/Au and Pt/Ti/Pt/Au gate were compared, and the thermal reliability instability was examined at elevated temperatures. There are many studies [26,27,28,29,30,31] on the reliability of the Ni and Pt gated structure, but our experimental method is unique and revived the property of Pt metal into its original condition.

## 2. Materials and Methods

The epitaxial layer structures were grown via the low-pressure metal-organic chemical vapor deposition (MOCVD) technique on 3-inch p-type Si wafers. This epitaxial structure consists of an Al_0.21_Ga_0.79_N barrier (28.5 nm), a Ga-polarity GaN channel layer (50 nm), and an AlGaN intermediate buffer layer (200 nm) atop the 3-inch p-Si substrate. Hall measurements revealed the mobility (*µ_n_Hall_*) and the sheet charge density (2DEG) to be 1300 cm^2^·V^−1^·s^−1^ and 9 × 10^12^ cm^−2^, respectively. The device fabrication involved mesa isolation etching, source/drain ohmic contact formation, and gate patterning. The mesa isolation etching was performed using a reactive ion etching (RIE) system; thereafter, the ohmic contacts were formed by standard Ti/Al/Ni/Au (25/160/40/100 nm) metallization on the source and drain regions, followed by rapid thermal annealing (RTA) at 830 °C for 30 s in ambient N_2_ to allow for the formation of the contacts on the AlGaN/GaN epi-structure. The contact resistance (R_c_) and sheet resistance (R_SH_) extracted by transmission-line-method (TLM) measurements were 1.2 Ω·mm and 320 Ω/□, respectively. Metallization was performed via the lift-off technique. The Schottky gate contacts were next patterned by photolithography; the Ni/Au (20/300 nm) and Pt/Ti/Pt/Au (8/20/20/300 nm) Schottky gate contacts were fabricated by e-beam evaporation, and Al_2_O_3_ (3 nm) was deposited as the surface passivation layer. A schematic cross-sectional structure and two different gate metals are shown in Figure 1a,b respectively. To understand the roles of the temperature and Schottky behavior characteristics, the electrical performance, I–V (current-voltage) characteristics, and C–V (capacitance–voltage) measurements were evaluated using a Keithley 4200SC semiconductor parameter analyzer and 4210-CVU, which were connected to a probe station with a temperature-controlled (Temptronic TP03000) heating plate.

The critical voltage was determined via incrementally stepped stress values of the V_G_ from −10 V, with the source and drain terminals grounded to avoid self-heating. At each stress step, similar gate length devices from each wafer were stressed for 1 min. To verify the degradation of the SBH under the off-state stress, a constant stress condition (V_D_ = 50 V, V_G_ = −7 V) was applied over a duration of 1 h to the gate and drain regions, with the source being grounded. To investigate the temperature dependence under the off-state stress, both devices, with the same gate length of Lg = 10 μm, were stressed at constant voltage (V_D_ = 50 V, V_G_ = −7 V) for 1 h by increasing the temperature from 298 K to 368 K in steps of 10 K. In our study, due to high gate leakage current, the calculation of the ideality factor did not provide satisfactory results for a gate length of 14 μm. Therefore, we considered a lower gate length of 10 μm to investigate the Schottky barrier degradation effect in both the temperature and stress related experiments. More than 35 devices with different gate lengths (10 μm, 11 μm, and 14 μm) were used in this study.

## 3. Results and Discussion

### 3.1. Impact of Schottky Contact Electrodes on Electrical Properties

Figure 2a–c shows the I–V characteristics of both the forward and reverse region, and Schottky characteristics (J–V characteristics) of the Ni/Au and Pt/Ti/Pt/Au Schottky contacts on the AlGaN/GaN HEMTs, respectively. In Figure 2a, the Ni/Au Schottky contact showed a forward current of 2.13 A/mm at 5 V while Pt/Ti/Pt/Au showed 0.5 A/mm. The high Schottky barrier height of the contact may cause a reduction in forward current [32]. In the reverse region (Figure 2b), the Pt/Ti/Pt/Au device degraded and showed higher leakage current 6.04 × 10^−5^ A/mm at −10 V than Ni/Au (5.29 × 10^−6^ A/mm). The SBHs and ideality factors for the Ni/Au and Pt/Ti/Pt/Au AlGaN/GaN HEMTs are given by (1) and (2), respectively:(1)$$J=J_{S}\left[\exp\left(\frac{qV}{nkT}\right)-1\right]$$
(2)$$J_{S}=A^{*}T^{2}\exp\left(\frac{-q\phi_{b}}{kT}\right)$$
where *J_S_* is the reverse saturation current density; n is the ideality factor; *A** is the effective Richardson constant; *T* is the absolute temperature; *ϕ_b_* is the SBH obtained from the saturation current density; and *k* is the Boltzmann constant [33]. The SBH of the Pt/Ti/Pt/Au contact at the reverse-biased region was observed to deteriorate, implying that the surface roughness caused by the high-energy Pt atoms deposited during e-beam evaporation process on the AlGaN/GaN HEMTs ultimately caused cracks in the MS contacts. This phenomenon can be attributed to the inhomogeneities at the MS interface and large deviations in the behaviors of the top electrodes despite the higher work function of Pt compared with Ni. Hence, the reverse leakage current of the Pt/Ti/Pt/Au Schottky contact was higher than that of the Ni/Au contact, as shown in Figure 2.

The capacitance–voltage (C–V) and conductance–voltage (G–V) characteristics of the Ni/Au and Pt/Ti/Pt/Au contacts on the AlGaN/GaN HEMTs were examined at 1 MHz, as shown in Figure 3. The threshold voltage of the Pt/Ti/Pt/Au Schottky contact in the forward region was observed to have a more positive shift of 0.28 V, as shown in Figure 3a. This means that the 2DEG carrier concentration below the Pt/Ti/Pt/Au Schottky contact was significantly reduced due to greater depletion under the Pt/Ti/Pt/Au contact than the Ni/Au contact, thereby indicating that a top electrode made of Pt has a higher work function than that made of Ni [7]. The conductance of the Pt-based Schottky contact was higher in the off region than that of the Ni/Au Schottky contact, indicating the degradation of the off-state leakage current.

The initial value (before stress) of the Schottky contact calculated both I–V and C–V methods. We measured the C–V characteristics of both devices at 1 MHz at room temperature (Figure 3a). The below equation demonstrated a Schottky barrier relation with capacitance characteristics [33].
(3)$$\frac{A^{2}}{C^{2}}=\frac{2\left(V_{bi}-\frac{kT}{q}-V\right)}{qN_{D}\varepsilon_{s}}$$
where *ϵ_s_* is the semiconductor permittivity; *V_bi_* is the built-in potential; *N_D_* is the doping concentration; *A* is the area; and *C* is the capacitance. In the reverse region, 1/*C*^2^ vs. *V* gave a straight line from which we calculated the flat band voltage *V*_0_ (intercept of x-axis) and built-in potential *V_bi_*. The barrier height can be calculated by the below expression,
(4)$$V_{n}=E_{C}-E_{F}=\frac{kT}{q}In\left(\frac{N_{C}}{N_{D}}\right)$$
where *E_C_* and *E_F_* represent the conduction band minima and Fermi energy level, respectively. We used a *N_C_* (effective density of states in the conduction band) of 1.7 × 10^18^ cm^−3^ [34]. Schottky barrier height can be expressed as
(5)$$\phi_{b}=V_{bi}+V_{n}+\frac{kT}{q}$$

From the C–V measurement, a Schottky barrier height value was obtained at 0.59 eV and 0.48 eV in Ni/Au and Pt/Ti/Pt/Au, respectively. Apparently, this was high compared to the I–V measured values [35] which is shown in Table 1.

The transconductance showed almost similar characteristics of both devices at the same gate length of 14 um. The drain current in the Pt/Ti/Pt/Au device in Figure 4b was slightly lower than the Ni/Au gated device, indicating that the 2DEG carrier concentration below the gate area had decreased [7]. The transfer characteristics of the Ni/Au and Pt/Ti/Pt/Au contacts on the AlGaN/GaN HEMTs are shown in Figure 4a. The off-state leakage current of the Pt/Ti/Pt/Au HEMT was slightly higher than that of the Ni/Au HEMT. In the voltage region below −1.5 V, the Pt-based Schottky gate yielded a high forward leakage current. Extra tunneling of the currents has been suggested in the Pt gate due to the surface level of the MS state, as demonstrated by Zhang et al. (2006) [36]. Thus, the leakage current increased instead of reducing at the minimum forward bias.

In Figure 5, the critical voltage of the Ni/Au HEMT, followed by a sudden increase in the gate leakage current, was about −25 V, which resulted in permanent defect sites at the MS interface. This sudden increase in the gate leakage current can be ascribed to the inverse piezoelectric effect [37]. In contrast, no sudden increase in the gate leakage current was observed up to −60 V in the Pt/Ti/Pt/Au HEMT [38]. Although all characteristics of the Pt/Ti/Pt/Au HEMT including Schottky diode, transistor I–V, and C–V measurements, showed significant improvements in the forward region, the device characteristics in the reverse region were degraded due to the inhomogeneities at the MS interface.

### 3.2. Reliability Instability Based on Temperature

To verify the degradation of the SBH from high electrical stress, the forward and reverse leakage currents of the Ni/Au and Pt/Ti/Pt/Au contacts on the AlGaN/GaN HEMTs were evaluated before and after off-state stress application (V_D_ = 50 V, V_G_ = −7 V) over a duration of 3600 s. The reverse leakage current of the Pt/Ti/Pt/Au contact after off-state stress application showed a greater reduction than that of the initial device while that of the Ni/Au increased, as shown in Figure 6. It is also interesting to note that the SBHs of the Ni/Au and Pt/Ti/Pt/Au after application of off-state stress decreased from 0.55 to 0.49 eV and increased from 0.46 to 0.69 eV, respectively. This means that the hot carriers under the off-state stress have a significantly effect on the MS. In fact, the stress condition at room temperature (25 °C) depends significantly on the gate voltage and electric field. The metallization schemes for the Schottky contacts on the AlGaN/GaN HEMT must thus be verified for thermal instabilities due to the Ga out-diffusion and Au interdiffusion at elevated temperatures.

The thermal reliability instabilities for the Ni/Au and Pt/Ti/Pt/Au HEMTs were examined in the temperature range of 298 to 368 K in intervals of 10 K. Figure 7 shows the J–V characteristics of the Ni/Au and Pt/Ti/Pt/Au contacts after application of off-state stress (V_D_ = 50 V, V_G_ = −7 V) at different temperatures. The reverse leakage currents of the Ni/Au HEMT before and after off-state stressing at 298 K were not degraded; in fact, the off-state stress with increasing temperature caused greater initial-parameter degradation rather than at room temperature [39], as shown in Figure 7a, which can easily generate more interface traps. In contrast, the reverse leakage currents of the Pt/Ti/Pt/Au HEMT decreased after off-state stressing at 298 K, with further reduction at 308 K under the same off-state stress conditions.

To investigate the barrier height and ideality factor of both devices, we repeated the same experiment with both Ni/Au and Pt/Ti/Pt/Au with different gate lengths of 10 μm and increased the temperature beyond 368 K to observe the degradation effect [40,41,42,43]. Figure 8a demonstrates the stress with temperature effect carried out on both devices. Barrier height of Ni/Au increased in a conventional way and the device burnt out at a 368 K temperature with the same off-state stress applied (V_D_ = 50 V, V_G_ = −7 V). The Pt/Ti/Pt/Au gated device had no breakdown issues after a temperature of 368 K. It was difficult to compare the ideality factor at a higher gate length (14 μm) due to high leakage current; therefore, we narrowed it down with shorter gate devices. We found a better ideality factor at a gate length of 10 μm compared to the gate length of 14 μm. Due to the high gate leakage current, the ideality factor (4.3) was high in Pt/Ti/Pt/Au [44,45]. However, after stress at 308 K, the ideality factor decreased, and was between 1 and 2 as the temperature increased.

Although the initial characteristics of the Pt/Ti/Pt/Au HEMT were determined by the inhomogeneities at the MS interface, the interfacial trap states can be reduced by curing the surface donor state of the Pt-based Schottky contact after off-state stressing. For an increase in temperature over 308 K under the same off-state stress condition, the reverse leakage current degradation is mitigated by the inverse piezoelectric effect mechanism. As the temperature increases further, the number of electrons with very high energies decreases. At increased temperature, greater phonon-induced carrier scattering mitigates carrier acceleration, which minimizes hot-carrier damage [46].

In other words, the number of moderate-energy carriers (0.5 eV to 2.5 eV) increases with an increase in temperature. Moderate-energy carriers could produce or reorganize flaws in the MS contacts after application of off-state stress if the defect activation energies are sufficiently low. Thus, it is necessary to analyze the concurrent SBH and hot-carrier degradation because of the interfacial trap states at the MS contacts.

## 4. Conclusions

Detailed reliability assessments and electrical characterizations were performed for Ni/Au and Pt/Ti/Pt/Au gate contacts on AlGaN/GaN HEMTs. Although the Pt-based gate was expected to have a high work function when used on top of the AlGaN, the SBH and reverse leakage current degraded in comparison with those of the Ni-based gate, implying that the high-energy Pt atoms induced surface roughness during the e-beam evaporation process. After applying off-state stresses and high temperatures, Pt/Ti/Pt/Au HEMTs showed abrupt reductions in the reverse leakage currents, which improved the SBH by suppressing the interfacial donor states. From the behaviors of the interfacial trap states of the MS contacts, the stability of the AlGaN/GaN HEMT could be ascertained by simultaneously investigating the SBH and reliability instability.

## Figures and Tables

**Figure 1 micromachines-13-00084-f001:**
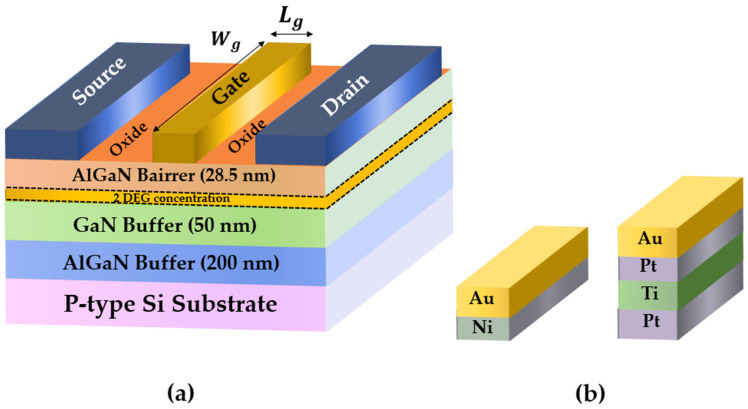
(**a**) Schematic cross section showing Ni/Au and Pt/Ti/Pt/Au gate contacts on the AlGaN/GaN HEMT. The Schottky gate contact on the AlGaN/GaN HEMT controls the current conduction in the GaN channel. (**b**) Gate metal stack for two different HEMTs.

**Figure 2 micromachines-13-00084-f002:**
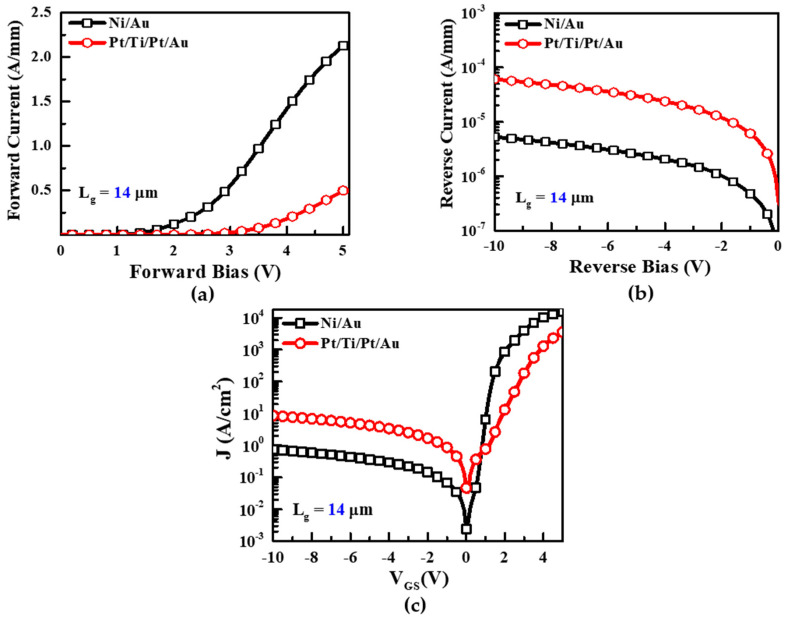
(**a**) Current–voltage plot (I–V) plot (characteristics) of the forward region and (**b**) reverse region. (**c**) Schottky characteristics of the contacts made of Ni/Au and Pt/Ti/Pt/Au fabricated on AlGaN/GaN HEMTs at room temperature.

**Figure 3 micromachines-13-00084-f003:**
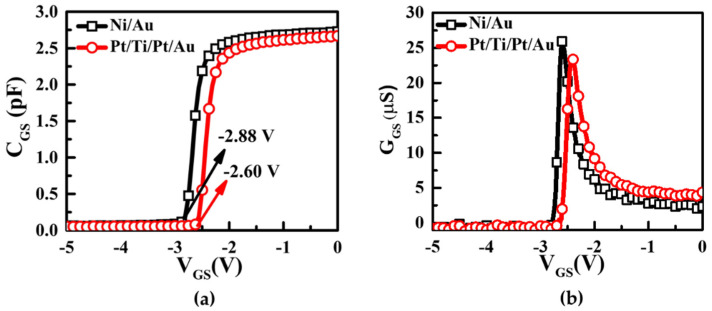
(**a**) Capacitance–voltage (C–V) and (**b**) conductance–voltage (G–V) characteristics of the Ni/Au and Pt/Ti/Pt/Au contacts on AlGaN/GaN HEMTs measured at 1 MHz; the Pt-based Schottky contact was observed to have a more positive shift.

**Figure 4 micromachines-13-00084-f004:**
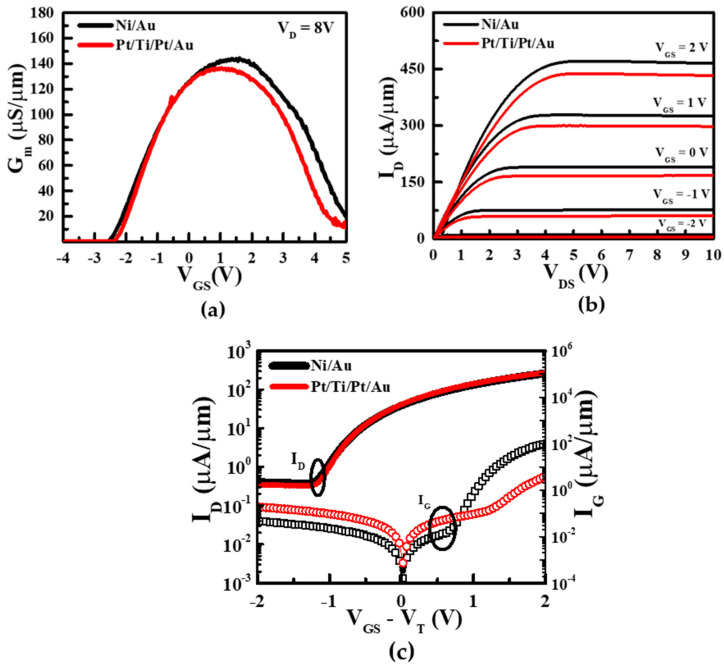
(**a**)Transconductance characteristics, (**b**) Output characteristics at different gate voltage and (**c**) transfer characteristics of Ni/Au and Pt/Ti/Pt/Au on AlGaN/GaN HEMTs with VD fixed at 8 V.

**Figure 5 micromachines-13-00084-f005:**
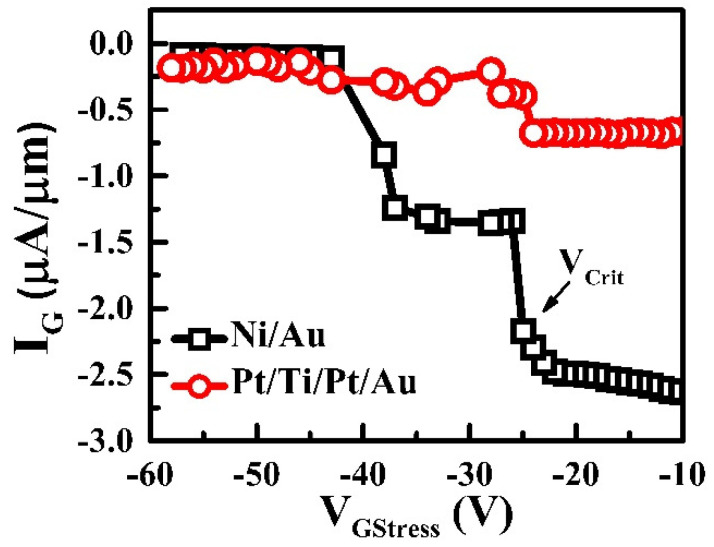
Critical voltages (V_crit_) of Ni/Au and Pt/Ti/Pt/Au on AlGaN/GaN HEMTs in the range of −10 to −60 V with stepped stresses. The V_crit_ of Ni/Au was about −25 V and that of Pt/Ti/Pt/Au was unspecified for up to −60 V.

**Figure 6 micromachines-13-00084-f006:**
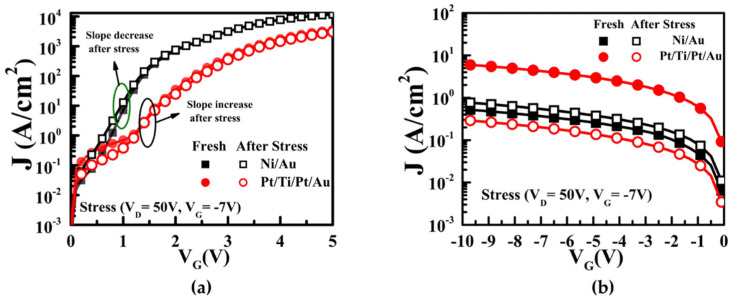
(**a**) Forward leakage current and (**b**) reverse leakage current of Ni/Au and Pt/Ti/Pt/Au on AlGaN/GaN HEMTs before and after off-state stress (V_D_ = 50 V, V_G_ = −7 V) during 3600 s.

**Figure 7 micromachines-13-00084-f007:**
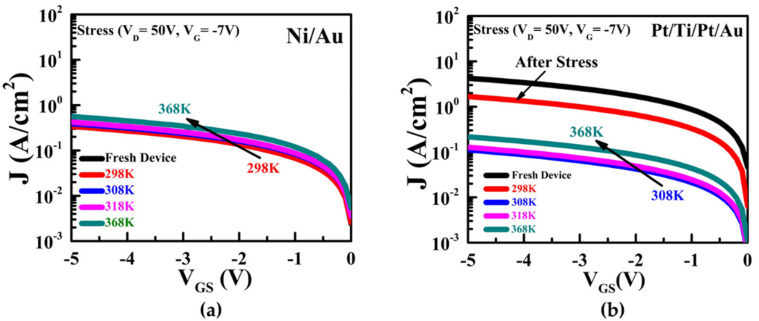
Reverse leakage current of the (**a**) Ni/Au and (**b**) Pt/Ti/Pt/Au on AlGaN/GaN HEMTs after off-state stress (V_D_ = 50 V, V_G_ = −7 V) in the temperature range of 298 K to 368 K measured at a temperature interval of 10 K. [Curves are not shown from 328 K to 358 K].

**Figure 8 micromachines-13-00084-f008:**
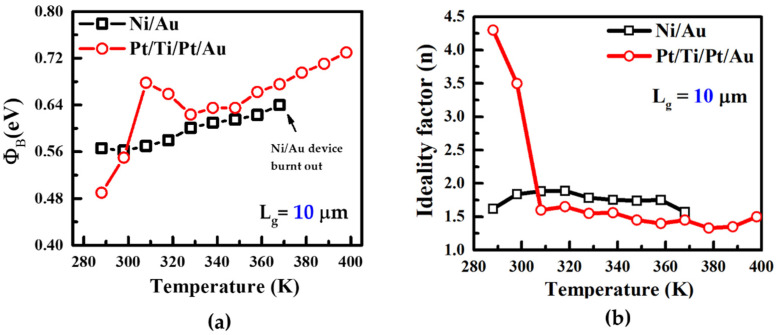
Barrier height (**a**) and ideality factor (**b**) of the Ni/Au and Pt/Ti/Pt/Au devices in the same experiment described in Figure 7. Ni/Au gated device burnt out after 378 K (**a**) and there was an increase in Schottky barrier height from 0.49 eV to 0.68 eV at 308 K in the Pt/Ti/Pt/Au gated device.

**Table 1 micromachines-13-00084-t001:** Comparison of I–V and C–V measured data of both devices at room temperature.

Schottky Barrier Height (*ϕ_b_*) (eV)	I–V Method	C–V Method
Ni/Au	0.55	0.59
Pt/Ti/Pt/Au	0.45	0.48

## Data Availability

The data presented in this study are available on request from the corresponding author. The data are not publicly available due to privacy issues.

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
