# Peer review of "Comprehensive Schottky Barrier Height Behavior and Reliability Instability with Ni/Au and Pt/Ti/Pt/Au on AlGaN/GaN High-Electron-Mobility Transistors"

_micromachines, 2022, doi:10.3390/mi13010084_

Round 1

Reviewer 1 Report

No further comments

Author Response

No further response for the Reviewer 1.

Reviewer 2 Report

Comment 1:

Authors claim in the introduction that their experimental technique is unique. In the method section could you please elaborate why your experiment is unique? From my point of perspective these are the typical fabrication and characterization techniques. Please be more specific about what differentiate your study from other studies [26-31]?

Comment 2:

Please add the unit at line 77  “320 Ω/□ respectively” under materials and methods

Comment 3:

Authors claim at lines 111-113 that “The SBH of the Pt/Ti/Pt/Au contact at the reverse-biased region was observed to deteriorate, implying that the surface roughness caused by the high-energy  Pt atoms deposited during E-beam evaporation process on the AlGaN/GaN HEMTs ultimately  caused cracks in the MS contacts.” If there is an SEM image indicating the cracks please provide it as supplementary file.

Comment 4:

Remove the comma after “ as shown in figure.” In line 188.

Comment 5:

Reference 3 and 20  are missing the name of the journal or editor

Comment 6:

I believe devices has reasonable performance

Comment 7:

In Figure 2. The font of the axis’s tittles and legends for 2a adn2b is same and 2c is different. It might be better if they are same.

Comment 8:

In figure 3b it might be better to use G in the y axis instead of cionductance to be uniform with figure 3a

Comment 9:

In the figure caption of figure 2 and 6 (a,b,c) are bolded but in figure 3,4,5,7, and 8  (a,b,c) are not bolded.

Comment 10:

Please verify that all the results shown here are for the same device with the same gate length of 14um. If not at any point please specify that.  I am asking this for your recently added results of transconductance characteristics. I realized later results are for 10 um devices please wrote these on the graph images as you stated in Figure 1. Also in the methods please mention how many device you use and what are the gate lengths of these devices.

Comment 11:

Between the lines of 89-95 you are using 3600 s and 1h. I would recommend you to be consistence to not to cause any confusion.

Comment 12:

Could you please comment on why SBH values from CV is higher than the one from IV?

Comment 13:

Overall please decide whether you will use past or present tense when you are explaining results. Please correct all the figures sometimes you wrote them as Figure sometimes figure sometimes fig. Please be consictent

Line 15: temperatures exceeding

Line 18: analyzing

Line 27: high-temperature, high-saturation

Line 28: Because of piezoelectric and spontaneous polarization

Line 44: the surface  donors using different gate metals

Line 45: GaN-based HEMTs is the improvement

Line 46: surfaces mediated

Line 48 : Pt, Au, 47 Ti, Ni, Au, and other highly

Line 53: The degradation of the hot carriers may cause a significant

Line 63: Pt gated structure, but our experimental

Line 70: remove the space before period.” respectively.” Have comma before respectively

Line 75: to allow the formation of the contacts

Line 81: A schematic cross-sectional structure and two different gate metal are shown in Figure 1(a) and 1(b), respectively

Line 98: Figure. 2 (a), (b), and (c) shows

Line 120:  as shown in the figure 3. Remove the comma after figure. I would recommend using short version of figure i.e fig.

Line 128: calculated both I-V and C-V methods

Line 129: C-V characteristics of both devics

Line 129: at 1 MHz at room temperature

Line 152: The transconductance characteristics show. Remove the “s” after “show”

Line 156: the Pt/Ti/Pt/Au HEMT is slightly 156 higher. Remove “ observed to be”

Line 173: “ Shows a greater

Line 177: “The hot carriers under the off-state stress significantly affect the MS

Line 178 “temperature (25°C) depends significantly on the

Line 198: In Figure 8 (a), there is

Line 202 It is difficult to compare ideality factors in higher gate length (14 μm) due to high 202 leakage current; therefore, we narrow down it with shorter gate devices

Line 203: We have found a better ideality factor in gate length

Line 204: compared to gate

Line 205: However, after stress at 308K

Line 206: as temperature increases

Line 196-Line 205 please check the grammar again

Line 209: For an increase in temperature

Line 224: After applying off-state

Line 224 during the E-beam

Author Response

Review 1, 30-December 2021

Dear Editor,

We are thankful for the constructive comments from the reviewers that we have considered in detail. After discussion and revision, we now submit this revised version of the manuscript to the MDPI Micromachines. In this letter, we enclose the point-by-point replies to all the valuable comments from the reviewers. We would appreciate your reconsideration of our work.

    ==========Author’s Responses to the Reviewer’s Comments=========

Reviewer 1

Reviewer wrote:

No comment.

Our response:

Thank you very much for carefully reviewing our manuscript.

Reviewer 2

Our response:

             Dear reviewer,

Thank you very much for carefully reviewing our manuscript and providing fruitful suggestions.  We appreciate the reviewer’s comment. We have made changes in our manuscript according to reviewer’s comment and observation.

Comment 1

Reviewer wrote:

  1. Authors claim in the introduction that their experimental technique is unique. In the method section could you please elaborate why your experiment is unique? From my point of perspective these are the typical fabrication and characterization techniques. Please be more specific about what differentiate your study from other studies [26-31]?

Our response:

Thank you very much for this outstanding question. All the mentioned studies, characterization technique based on only temperature or only stressed related issues applied on the device separately. We emphasized both temperature and stress related issues together and characterized those device performances accordingly. For this reason, we think that our method is clearly distinguishable from other studies. 

Corresponding change in manuscript: No.

Comment 2

Reviewer wrote:

  1. Please add the unit at line 77  “320 Ω/□ respectively” under materials and methods

Our response:

Thank you for pointing out this issue.  We have corrected it in our revised manuscript.

Corresponding change in manuscript: Yes. Highlighted in yellow in the main manuscript.

Location of change:

             Section: 2. Materials and Method

             Page- 2.  Line no 76.

Comment 3

Reviewer wrote:

  1. Authors claim at lines 111-113 that “The SBH of the Pt/Ti/Pt/Au contact at the reverse-biased region was observed to deteriorate, implying that the surface roughness caused by the high-energy  Pt atoms deposited during E-beam evaporation process on the AlGaN/GaN HEMTs ultimately  caused cracks in the MS contacts.” If there is an SEM image indicating the cracks, please provide it as supplementary file.

Our response:

Thank you for your valuable comment.  Currently we don’t have SEM image indicating the cracks, but we will consider this in our future work.

Corresponding change in manuscript: No.

Comment 4

Reviewer wrote:

  1. Remove the comma after “ as shown in figure.” In line 188

Our response:

Thank you for your valuable comment. We have corrected it in our revised manuscript.

Corresponding change in manuscript: Yes. Highlighted in yellow in the main manuscript.

Location of change:

          Section: 3. Result and Discussion  

         Sub-section:3.1 Impact of Schottky contact electrodes on electrical properties.

         Page-7 and new line no : 191

Comment 5

Reviewer wrote:

  1. Reference 3 and 20  are missing the name of the journal or editor.

Our response:

Thank you for your valuable comment. We have corrected it in our revised manuscript.

Corresponding change in manuscript: Yes. Highlighted in yellow in the main manuscript.

Location of change:

Section: References

Reference No : 03,  corrected the authors name, Journal name and “doi” number. Line     No: 247-248.

Reference No : 20,  corrected the authors name, Journal name and “doi” number. Line No: 289-290.

Comment 6

Reviewer wrote:

  1. I believe devices has reasonable performance.

Our response:

Thank you for your valuable comment and thank you for understanding our devices    performances.

Comment 7

Reviewer wrote:

  1. In Figure 2. The font of the axis’s tittles and legends for 2a and 2b is same and 2c is different. It might be better if they are same.

Our response:

Thank you for your valuable comment. We have corrected it in our revised manuscript.

Corresponding change in manuscript: Yes. Highlighted in yellow in the main manuscript.

Location of change:

Section: 2. Materials and Methods

             Figure : 2 (c) changed its legend font and axis titles.

Comment 8

Reviewer wrote:

  1. In figure 3b it might be better to use G in the y axis instead of conductance to be uniform with figure 3a.

Our response:

Thank you for your valuable comment. We have corrected it in our revised manuscript.

Corresponding change in manuscript: Yes. Highlighted in yellow in the main manuscript.

Location of change:

Section: 3. Result and Discussion

Sub-section: 3.1 Impact of Schottky contact electrodes on electrical properties.

             Figure : 3 (b) changed G in the y axis instead of conductance.

Comment 9

Reviewer wrote:

  1. In the figure caption of figure 2 and 6 (a,b,c) are bolded but in figure 3,4,5,7, and 8  (a,b,c) are not bolded.

Our response:

Thank you for your valuable comment. We have corrected it in our revised manuscript.

Corresponding change in manuscript: Yes. Highlighted in yellow in the main manuscript.

Location of change:

  1. Section: 3. Result and Discussion

Sub-section: 3.1 Impact of Schottky contact electrodes on electrical properties.

             Figure caption : 3 (a) and (b) are bolded. 

             Figure caption : 4 (a) and (b) are bolded. 

Figure caption : 6 (a) and (b) are bolded. 

  1. Section: 3. Result and Discussion

Sub-section: 3.2 Reliability instability based on temperature.

Figure caption : 7 (a) and (b) are bolded. 

Comment 10

Reviewer wrote:

  1. Please verify that all the results shown here are for the same device with the same gate length of 14um. If not at any point please specify that.  I am asking this for your recently added results of transconductance characteristics. I realized later results are for 10 um devices please wrote these on the graph images as you stated in Figure 1. Also, in the methods please mention how many devices you use and what are the gate lengths of these devices.

Our response:

Thank you for your valuable comment. Our transconductance characteristics is the same gate length 14 um. We have corrected it at the line no: 155. We have mentioned the reason for selecting lower gate length at the line no: 94 to 98.

Corresponding change in manuscript: Yes. Highlighted in yellow in the main manuscript.

Location of change:

  1. Section: 2. Materials and Method

             Line no: 94 to 97.

  1. Section: 3. Result and Discussion

Sub-section: 3.1 Impact of Schottky contact electrodes on electrical properties.

Line no: 155

Comment 11

Reviewer wrote:

  1. Between the lines of 89-95 you are using 3600 s and 1h. I would recommend you to be consistence to not to cause any confusion.

Our response:

Thank you for your valuable comment. We have corrected it.

Corresponding change in manuscript: Yes. Highlighted in yellow in the main manuscript.

Location of change:

Section: 2. Materials and Method

             Line no: 90. Corrected it into 1 h.

Comment 12

Reviewer wrote:

  1. Could you please comment on why SBH values from CV is higher than the one from IV?

Our response:

Thank you for your valuable comment. There are many explanations for this, but no one is satisfactory answer yet. We think that charges trapped between the metal and semiconductor Schottky diode which may required to correct the applied voltage.

There is flat band voltage ( ) in the main equation,  which is added in the C-V measurement. We think that extraction of the  results high Schottky barrier height calculation.

Corresponding change in manuscript: No.

Comment 13

Reviewer wrote:

  1. Overall, please decide whether you will use past or present tense when you are explaining results. Please correct all the figures sometimes you wrote them as Figure sometimes figure sometimes fig. Please be consistent.

Our response:

Thank you for your valuable comment. We have corrected all the mentioned lines in our revised manuscript. ‘figures’ also corrected into ‘Figure’.

Corresponding change in manuscript: Yes, below comments and grammar related issues.

Line 15: temperatures exceeding

Line 18: analyzing

Line 27: high-temperature, high-saturation

Line 28: Because of piezoelectric and spontaneous polarization

Line 44: the surface  donors using different gate metals

Line 45: GaN-based HEMTs is the improvement

Line 46: surfaces mediated

Line 48 : Pt, Au, 47 Ti, Ni, Au, and other highly.

             We don’t understand this metal name. ‘47 Ti’

Line 53: The degradation of the hot carriers may cause a significant

Line 63: Pt gated structure, but our experimental

Line 70: remove the space before period.” respectively.” Have comma before respectively

Line 75: to allow the formation of the contacts

Line 81: A schematic cross-sectional structure and two different gate metal are shown in Figure 1(a) and 1(b), respectively

Line 98: Figure. 2 (a), (b), and (c) shows

Line 120:  as shown in the figure 3. Remove the comma after figure. I would recommend using short version of figure i.e fig.

Line 128: calculated both I-V and C-V methods

Line 129: C-V characteristics of both devics

Line 129: at 1 MHz at room temperature

Line 152: The transconductance characteristics show. Remove the “s” after “show”

Line 156: the Pt/Ti/Pt/Au HEMT is slightly 156 higher. Remove “ observed to be”

Line 173: “ Shows greater

Line 177: “The hot carriers under the off-state stress significantly affect the MS

Line 178 “temperature (25°C) depends significantly on the

Line 198: In Figure 8 (a), there is

Line 202 It is difficult to compare ideality factors in higher gate length (14 μm) due to high 202 leakage currenttherefore, we narrow down it with shorter gate devices

Line 203: We have found a better ideality factor in gate length

Line 204: compared to gate

Line 205: However, after stress at 308K

Line 206: as temperature increases

Line 196-Line 205 please check the grammar again

Sentence and grammar changed from past particle to past tense.

Line 209: For an increase in temperature

Line 224: After applying off-state

Line 224 during the E-beam

This manuscript is a resubmission of an earlier submission. The following is a list of the peer review reports and author responses from that submission.

Round 1

Reviewer 1 Report

Though this is an interesting work for those who are working in the field of III-Nitride materials and devices, but the present form may not be good enough for publication.

1) 2DEG properties of the AlGaN/GaN HEMT are missing.  Authors should provide those values (e.g. Sheet resistance, 2DEG Mobility and 2DEG sheet carrier density) in the revised manuscript. 

2) In the Figure 1 (b), authors have stated the metal stack sequence in a reverse order.  it is common that the Ni or Pt is in contact with AlGaN (not Au is touching the GaN). This has to be corrected in the revised manuscript.

3) The Schottky properties of two types of gate stack is not very good. The reverse leakage current is also very high. Authors should provide the measured Schottky properties for both metal stack. Authors should also provide the Schottky barrier height values for both types of measurements (I-V and C-V).

4) It is common that Pt-based Schottky properties are better than the Ni-based Schottky properties.  To be make sure that Authors should provide the barrier height, ideality factor values for these two types of metal stack based Schottky gate.  Authors should also benchmark with prior arts in a graphical or in a table form. This topic is well studied in the literature.  

5) Page #4, Line #122: Authors stated "The off-state leakage current of the Pt/Ti/Pt/Au HEMT is observed to be slightly lower than that of the Ni/Au HEMT".   It is supposed to be slightly higher than that of Ni/Au HEMT.  Authors should correct it in the revised manuscript.

6)  Authors are claiming that the reverse piezoelectric effect is dominated in the Ni-based Schottky gate when it compared with Pt-based Schottky gate.  Authors has to provide a solid supporting data (instead of citing only a reference).  This is the main objective of this work.  Authors should provide a solid evidence (e.g. Do the Cross-sectional transmission electron microscopy or delaying the metal and see under the gate).

Author Response

Reviewer wrote:

Though this is an interesting work for those who are working in the field of III-Nitride materials and devices, but the present form may not be good enough for publication.

Our response:

Thank you very much for carefully reviewing our manuscript and providing fruitful suggestions. We appreciate the reviewer’s comment.

We have made changes in our manuscript according to reviewer’s comment and observation. We have tried to explain all the possible facts and reason.

Comment 1

Reviewer wrote:

  1. 2-DEG properties of the AlGaN/GaN HEMT are missing. Authors should provide those values (e.g., Sheet resistance, 2DEG Mobility and 2DEG sheet carrier density) in the revised manuscript.

Our response:

Thank you very much for pointing out of this issue. We have mentioned 2DEG properties of the AlGaN/GaN HEMT.

Corresponding change in manuscript: Yes, We have added the properties in the main manuscript.

Hall measurements revealed the mobility (µn_Hall) and the sheet charge density (2DEG) to be 1300 cm2·V−1·s−1 and 9 × 1012 cm−2, respectively

And “The contact resistance (Rc) and sheet resistance (RSH) were extracted by transmission-line-method (TLM) measurements to be 1.2 Ω·mm and 320 Ω/□ respectively.”

Location of change:

             Section: 2. Materials and method.

             Page- 2. Line no 69 and 75.  Highlighted in yellow in the main manuscript.

Comment 2

Reviewer wrote:

  1. In the Figure 1 (b), authors have stated the metal stack sequence in a reverse order. it is common that the Ni or Pt is in contact with AlGaN (not Au is touching the GaN). This has to be corrected in the revised manuscript

Our response:

Thank you very much for pointing out of this issue. We have corrected it in our manuscript.

Corresponding change in manuscript: Yes, We have changed the figure 1(b) and correct the metal stacks.

Location of change:

             Section: 2. Materials and Method

             Page- 2.  Figure 1 (b).  Highlighted in yellow in the main manuscript.

Comment 3

Reviewer wrote:

  1. The Schottky properties of two types of gate stack is not very good. The reverse leakage current is also very high. Authors should provide the measured Schottky properties for both metal stacks. Authors should also provide the Schottky barrier height values for both types of measurements (I-V and C-V).

Our response:

Thank you very much for pointing out of this issue. We have added two figure (2 (a), 2(b)) for visualization of the properties of both devices. And added one reference.

We have added  C-V measurement method and compare the result in table no. 1. It is apparent that Schottky barrier is slightly higher in C-V measurement compared to I-V method. We have measured fresh device.

Corresponding change in manuscript: Yes, we have added two figures to compare the properties of both devices (1st change). Yes, we have written C-V measurement equations and compare the result in table 1. (2nd change).

Location of change:

1st Change:

Section: 2. Materials and Method.

             Page- 3. Line No: 100 to 104 .Highlighted in yellow in the main manuscript.

             Added Reference : No. 32

2nd Change:

Section: 3. Results and Discussion

Sub-section: 3.1 Impact of Schottky contact electrodes on electrical properties.

             Page- 4 & 5. Line No: 128 to 146 .Highlighted in yellow in the main manuscript.

Table added : Table no. 1 Comparison of I-V and C-V measured Data of both devices at room temperature.

Comment 4

Reviewer wrote:

  1. It is common that Pt-based Schottky properties are better than the Ni-based Schottky properties. To be make sure that Authors should provide the barrier height, ideality factor values for these two types of metal stack based Schottky gate. Authors should also benchmark with prior arts in a graphical or in a table form. This topic is well studied in the literature.

Our response:

Thank you very much for pointing out of this issue. This is very important question. For this issue, we added barrier height behavior and ideality factor behavior in Figure no. 8. in a graphical way.

Corresponding change in manuscript: Yes, we have added the behavior of barrier height and ideality factor in figure no 8.

Location of change:

             Section: 3. Results and Discussion

Sub-section: 3.2 Reliability instability based on temperature.

             Page- 7 and 8. Line No: 191 to 203 .Highlighted in yellow in the main manuscript.

New figure added : Figure No. 8

Comment 5

Reviewer wrote:

  1. Page #4, Line #122: Authors stated, "The off-state leakage current of the Pt/Ti/Pt/Au HEMT is observed to be slightly lower than that of the Ni/Au HEMT".   It is supposed to be slightly higher than that of Ni/Au HEMT. Authors should correct it in the revised manuscript.

Our response:

Thank you very much for pointing out of this issue. We have corrected this in our manuscript.

Location of change:

             Section: 3. Results and Discussion

Sub-section: 3.1 Impact of Schottky contact electrodes on electrical properties.

             Page- 5 . Line No: 152 .Highlighted in yellow in the main manuscript.

Comment 6

Reviewer wrote:

  1. Authors are claiming that the reverse piezoelectric effect is dominated in the Ni-based Schottky gate when it compared with Pt-based Schottky gate.  Authors has to provide a solid supporting data (instead of citing only a reference).  This is the main objective of this work.  Authors should provide solid evidence (e.g. Do the Cross-sectional transmission electron microscopy or delaying the metal and see under the gate).

Our response:

Thank you very much for pointing out of this issue. Your comment is highly appreciated. In our case, we currently cannot do this kind of experiment, but this is very important to see the internal behavior. We already considered it priority basis and included  in our future work.

Reviewer 2 Report

In this manuscript, the authors have systematically researched the Schottky barrier height behavior and off-state stress temperature-dependent stability of Ni/Au and Pt/Ti/Pt/Au gate contacts on AlGaN/GaN high-electron-mobility transistors (HEMTs). The performance degradation is mainly affected by the interfacial trap states of the MS contacts. There are still some concerns and details that need to be clarified and modified before the consideration for publication.

  1. What research objectives do the authors want to achieve through this study, is there a corresponding clear conclusion? In other words, what is the main significance and innovation of this article, and what guidance does it have for the preparation of GaN-based Schottky structures in the future?
  2. The interface of the metal-semiconductor should be Ni/AlGaN and Pt/AlGaN. Thus, Figure 1(b) is suggested to be modified to avoid misunderstanding. Additionally, please note the border issues in other images.
  3. Please check the correctness of formula 1.
  4. The author explained that the inhomogeneity and large deviations in the behaviors of the Pt/Ti/Pt/Au gate led to an increase in the reverse leakage current of the device, indicating that the influence of the large metal work function of Pt is not obvious. Why is the 2DEG concentration of the Pt gate device further depleted? (The author has attributed it to the large work function of the Pt gate.)
  5. Why does the SBH of the Pt/Ti/Pt/Au device increase from 0.46 to 0.69eV after off-state stress applied?

Author Response

Reviewer wrote:

In this manuscript, the authors have systematically researched the Schottky barrier height behavior and off-state stress temperature-dependent stability of Ni/Au and Pt/Ti/Pt/Au gate contacts on AlGaN/GaN high-electron-mobility transistors (HEMTs).  The performance degradation is mainly affected by the interfacial trap states of the MS contacts.  There are still some concerns and details that need to be clarified and modified before the consideration for publication.

Our response:

             Dear reviewer,

Thank you very much for carefully reviewing our manuscript and providing fruitful suggestions.  We appreciate the reviewer’s comment.

We have made changes in our manuscript according to reviewer’s comment and observation. We have tried to explain the facts and reason why Schottky barrier height is increased after off-state stress in Pt/Ti/Pt/Au gated device.

Comment 1

Reviewer wrote:

  1. What research objectives do the authors want to achieve through this study, is there a corresponding clear conclusion? In other words, what is the main significance and innovation of this article, and what guidance does it have for the preparation of GaN-based Schottky structures in the future?

Our response:

Thank you very much for this outstanding question.  We have demonstrated that due to surface roughness and interfacial traps Pt gated device showed lower Schottky barrier height because the device fabricated through E-beam evaporation.  Even though Pt gated device should show high Schottky barrier height owe to high work function, but our experiment depicted completely different phenomenon in the reverse region.  Pt reacts with Ga to form gallides, which deteriorate its performance. Due to the high diffusivity of some gases in Pt, these connections were found to be particularly sensitive to surface conditions and prone to the development of bubbles during E-beam evaporation. Much research showed that Pt gated device could be superior candidate for AlGaN/GaN HEMT rather than Ni/Au but there is still concern about its instability on AlGaN surface due to interfacial sensitive trap state on metal semiconductor (MS) surface (reverse region) which could be our possible explanation and conclusion in this study.

For future suggestion, as we mentioned earlier that Pt gated structed suffers from poor metal semiconductor (MS) interface during E-beam evaporation.  But in reliability perspective, Pt gated device showed better performance if the surface preparation ( AlGaN barrier) can be done with great attention.  We will consider this in our future work.

Corresponding change in manuscript: No.

Comment 2

Reviewer wrote:

  1. The interface of the metal-semiconductor should be Ni/AlGaN and Pt/AlGaN. Thus, Figure 1(b) is suggested to be modified to avoid misunderstanding. Additionally, please note the border issues in other images.

Our response:

Thank you for pointing out this issue.  We have corrected it in our manuscript.

Corresponding change in manuscript: Yes, We changed the figure 1(b) and correct the border issue in figure 1(a).

Location of change:

             Section: 2. Materials and Method

             Page- 2.  Figure 1 .  Highlighted in yellow in the main manuscript.

Comment 3

Reviewer wrote:

  1. Please check the correctness of formula 1.

Our response:

Thank you for your valuable comment.  We have corrected it according to your suggestion.

Corresponding change in manuscript: Yes, We have corrected our formula.

Location of change:

             Section: Result and Discussion

             Page-4 and line no 106 . Highlighted in yellow in the main manuscript.

Comment 4

Reviewer wrote:

  1. The author explained that the inhomogeneity and large deviations in the behaviors of the Pt/Ti/Pt/Au gate led to an increase in the reverse leakage current of the device, indicating that the influence of the large metal work function of Pt is not obvious. Why is the 2DEG concentration of the Pt gate device further depleted?  (The author has attributed it to the large work function of the Pt gate.)

Our response:

Thank you for your valuable comment.  From C-V (Figure 3. (b)) and G-V (Figure 3. (b)) shows that positive threshold voltage shift in Pt/Ti/Pt/Au contact indicating 2 DEG concentration depleted due to high work function in forward region.  But  in reverse region, leakage current increases.  We have highlighted gate leakage current mainly which degraded device performance in reverse region.

Corresponding change in manuscript: Yes, we have added one reference for clearing the fact of metal work function.

Location of change:

          Section: Result and Discussion  

         Page-4 and Line no. 122- 125 . Highlighted in yellow in the main manuscript.

        Added Reference no. 7

Comment 5

Reviewer wrote:

  1. Why does the SBH of the Pt/Ti/Pt/Au device increase from 0.46 to 0.69eV after off-state stress applied?

Our response:

Thank you for your valuable comment. We have pointed out at line no 157 to 159 that the initial characteristics of the Pt/Ti/Pt/Au HEMT are determined by the inhomogeneities at the MS interface, the interfacial trap states can be reduced by curing the surface donor state of the Pt-based Schottky contact after off-state stressing.  And in line no 166, we have pointed out that moderate-energy carriers could produce or reorganize flaws in the MS contacts after application of off-state stress if the defect activation energies are sufficiently low. Due to suppression of interfacial trap after off-state stress, Schottky barrier height increase from 0.46 to 0.69 eV. We have included figure no. 8 for better view of barrier height increase and ideality factor decrease issue in both devices.

Corresponding change in manuscript:

Section: 3. Results and Discussion

Sub-section: 3.2 Reliability instability based on temperature.

Page- 7 and 8. Line No: 191 to 203 .Highlighted in yellow in the main manuscript.

New figure added : Figure No. 8

Round 2

Reviewer 1 Report

After seeing the ideality factor values, the performance of the devices are not very good.  It is hard to compare the results between two different types of the metal stacks.  

For example:  The ideality factor of the Schottky gate is poor.  Normally, the Ideality factor should be lower than 2.0 (max).  The room temperature Ideality factor is more than 4.0.